# A Fatal Case of Native Valve Endocarditis with Multiple Embolic Phenomena and Invasive Methicillin-Resistant *Staphylococcus aureus* Bacteremia: A Case Report from the Maldives

**DOI:** 10.3390/tropicalmed8010053

**Published:** 2023-01-10

**Authors:** Ali Shafeeq, Hisham Ahmed Imad, Ahmed Azhad, Migdhaadh Shareef, Mohamed Shaneez Najmy, Mohamed Mausool Siraj, Mohamed Sunil, Rimsha Rafeeu, Aishath Sofa Moosa, Ahmed Shaheed, Thundon Ngamprasertchai, Wasin Matsee, Pyae Linn Aung, Wang Nguitragool, Tatsuo Shioda

**Affiliations:** 1National Cardiac Center, Indira Gandhi Memorial Hospital, Malé 20002, Maldives; 2Mahidol Vivax Research Unit, Faculty of Tropical Medicine, Mahidol University, Bangkok 10400, Thailand; 3Thai Travel Clinic, Hospital for Tropical Diseases, Bangkok 10400, Thailand; 4Center for Infectious Disease Education and Research, Department of Viral Infections, Research Institute for Microbial Diseases, Osaka University, Osaka 565-0871, Japan; 5Trauma and Emergency, Indira Gandhi Memorial Hospital, Malé 20002, Maldives; 6Department of Medicine, Indira Gandhi Memorial Hospital, Malé 20002, Maldives; 7Department of Clinical Tropical Medicine, Faculty of Tropical Medicine, Mahidol University, Bangkok 10400, Thailand; 8Department of Molecular Tropical Medicine and Genetics, Faculty of Tropical Medicine, Mahidol University, Bangkok 10400, Thailand

**Keywords:** infective endocarditis, methicillin-resistant *Staphylococcus aureus*, mitral valve

## Abstract

Infective endocarditis (IE) is a life-threatening condition caused by infection within the endocardium of the heart and commonly involves the valves. The subsequent cascading inflammation leads to the appearance of a highly friable thrombus that is large enough to become lodged within the heart chambers. As a result, fever, fatigue, heart murmurs, and embolization phenomena may be seen in patients with IE. Embolization results in the seeding of bacteria and obstruction of circulation, causing cell ischemia. Of concern, bacteria with the potential to gain pan-drug resistance, such as methicillin-resistant *Staphylococcus aureus* (MRSA), are increasingly being identified as the causative agent of IE in hospitals and among intravenous drug abusers. We retrospectively reviewed de-identified clinical data to summarize the clinical course of a patient with MRSA isolated using an automated blood culture system. At the time of presentation, the patient showed a poor consciousness level, and the calculated Glasgow scale was 10/15. A high-grade fever with circulatory shock indicated an occult infection, and a systolic murmur was observed with peripheral signs of embolization. This case demonstrated the emerging threat of antimicrobial resistance in the community and revealed clinical findings of IE that may be helpful to clinicians for the early recognition of the disease. The management of such cases requires a multi-specialty approach, which is not widely available in small-island developing states such as the Maldives.

## 1. Introduction

Before the advent of antimicrobial therapy, infectious diseases were the leading cause of death in all age groups [1]. Nevertheless, with contemporary advancements in medical science, infectious diseases now rank below cardio-cerebrovascular diseases, accidents, and malignancies as the top causes of death worldwide [2]. However, the mortality rate still remains high for certain infectious diseases, such as infective endocarditis (IE) [3].

When IE presents acutely, it is a life-threatening and complex medical emergency that is inevitably fatal without prompt treatment with broad-spectrum antimicrobials [4] in addition to early integral surgical correction of the abnormal excrescence appearing on heart valves [5]. Delayed treatment and management complicate the clinical course and may result in extra-cardiac embolization, which has been reported to occur in a quarter of patients with IE [6,7]. The spectrum of clinical manifestations in IE includes the eruption of non-tender hemorrhagic macular purpuric lesions over most distal peripheral sites of the hands and feet, including the appearance of hemorrhage streaks underneath the nails, which result from a vascular phenomenon that transpires in IE. Additionally, erythema over the finger pulps and hematuria may be observed from the deposition of circulating immune complexes, which may also initiate glomerulonephritis. Furthermore, septic seeding due to the formation of breakaway emboli of fibrin-platelet plugs containing bacteria may occur to multiple organ systems, including within the eye, causing surrounding retinal hemorrhage [8].

Since 1885, IE has been recognized as a pathological entity of an infective process [9]. Some forerunners such as Sir William Osler, Edward Janeway, and Mortiz Roth have had pathognomonic findings of IE named after them [10]. Presently, the incidence of IE is increasing in different regions, and IE has been reported to affect up to 7 to 10 people per 100,000 person-years [11]. Several predisposing factors for the development of IE have shifted from conditions such as rheumatic heart disease or congenital heart diseases to degenerative valve disease due to the increasingly aging population [12,13]. In addition, the increased use of prosthesis and intra-cardiac devices are factors that predispose an individual to the development of IE [14]. Individuals who are severely immunocompromised or those with a prior history of endocarditis are also considered to be at risk of developing IE [15].

Although the pathogenesis of IE has not yet been fully elucidated, it was postulated that it is initiated by the activation of endocardial endothelial cells by an antigen or denudation from contaminants introduced into the circulatory system. The inflicted injury to the endothelial cell lining frequently involves the endothelial layer of the heart valves. Endotheliitis initiates a hyper-coagulant state via an enzymatic process, depositing fibrin and webbing the anterior surfaces of the heart valvular structures (cusps and leaflets) and further trapping platelets and other blood cells and components of blood in the site [16]. This causes blood flow turbulence within the heart chambers, promoting the ongoing endothelial injury [17]. Concurrently, the translocated bacterium adheres to the site, triggering an inflammasome response and up-regulation of pro-inflammatory cytokines [18]. Additionally, a pedunculated mass is formed by the snowballing effect of the vegetative excrescence within the heart chambers. In large vegetation, visualization of its oscillative motion is possible by transthoracic echocardiography [19]. Two approaches are utilized for optimizing the field of view, i.e., a transthoracic approach and a transesophageal approach; the latter has been reported to be more sensitive in identifying vegetation than the less-invasive transthoracic echocardiography. The soft vegetative excrescence is highly friable and can break loose into the circulatory system as an embolus that can subsequently obstruct the distal microvascular circulatory system and seed bacteria into other organs [20].

Here, we describe a severe complex medical condition requiring a multi-disciplinary approach for effective treatment and management required in order to prevent the patient from succumbing to illness. Further, we highlight the alarming discovery of multi-drug-resistant bacteria such as methicillin-resistant *Staphylococcus aureus* (MRSA) within the community in the Maldives, and bring forth the worsening of social issues and behaviors that risk acquiring and transmitting pathogens within certain groups of individuals in the Maldivian population. Additionally, this study will showcase important clinical findings that are pathognomonic to infective endocarditis, which we anticipate will be highly useful for clinicians and medical students.

## 2. Materials and Methods

The present case was referred from a peripheral hospital in the Maldives to Indira Gandhi Memorial Hospital (IGMH), Malé, the Maldives, in April 2022. The patient was subsequently admitted to the National Cardiac Center with the diagnosis of native valve IE.

### 2.1. Clinical and Laboratory Data

De-identified clinical and laboratory data were retrospectively reviewed to summarize the clinical course of the presented case. Blood culture and susceptibility testing were performed using an automated culture system (bioMerieux, Durham, NC, USA). Additional bloodborne pathogen screening was performed using an electrochemiluminescence immunoassay (ECLIA; Roche, Mannheim, Germany) for hepatitis B surface (HBs) antigen and anti-hepatitis HBs antibody, (Humasis, Gyeonggi-do, Republic of Korea) for anti-HCV antibody, ECLIA (Roche) for HIV antibody/antigen, and RT-PCR (Liferiver, San Diego, CA, USA) for SARS-CoV-2.

### 2.2. Literature Review of Cases with Native Valve IE with MRSA in IVDU

We searched *Google Scholar* to match the presented case with other similar IE cases. A total of five keywords (native valve; infective endocarditis; MRSA; IVDU; septic emboli; and case reports) along with a three-year period were used. Subsequently, 50 papers published in the English language were retrieved for further scanning. After reading the abstracts, we first excluded 46 papers. One was the current study in pre-print form; one was duplicated; ten were either general reviews or basic literature or book chapters for basic etiology, diagnosis, and treatment of IE; eleven were analyses of a case series either in retrospective or prospective approach; three case reports provided the information regarding IE with MSSA; and the remaining twenty were not relevant to the presented case, as they described different areas of interest such as bioengineering, cardiac magnetic resonance imaging, or general infectious diseases. This exclusion left a total of four papers to be included in a detailed review. Among them, another two were repudiated because one paper lacked the details of the patient’s presentation and cardiovascular-related findings, and another reported a patient with underlying malignancy and ongoing treatment for many years. Consequently, two final case reports were eventually included in our review, and we tabulated the findings for easy comparison in the Appendix A.

## 3. Case Description

A 31-year-old male with a provisional diagnosis of sepsis was promptly medically evacuated from a primary health center to the emergency department at a central referral tertiary care hospital (IGMH) in the Maldives. At the time of arrival to the emergency department, the patient was drowsy, was observed to show a poor response to pain stimulus, and exhibited peripheral cyanosis. Anisocoria was present, and the pupils did not react to light stimulus. Ptosis of the right eye was seen. The pupils measured 3 mm in the right eye and 2 mm in the left eye. The palpebral conjunctivae appeared pale without conjunctival hemorrhage. Furthermore, there was no neck stiffness, signs of meningeal irritation, or evidence of cervical lymphadenopathy. No jaundice, edema, or bleeding was apparent during the physical examination. Additionally, no abnormality was observed within the oral cavity, and the posterior pharyngeal wall, tonsils, and gag reflex appeared to be normal. No suspicious rash, localized ecchymosis over the trunk, or other visible physical abnormality on the body was observed except for obvious multiple track marks over the right cubital fossa in addition to clubbing of the terminal phalanges, Janeway lesions, and splinter hemorrhages, as shown in Figure 1.

The recorded vital signs at presentation included a biometric thermal body temperature of 38.1 °C, a pulse rate of 142 beats per minute, and a blood pressure reading of 88/58 mmHg, with a prolonged capillary refilling time (>2 s) and a respiratory rate of 52 breaths per min.

Examination of the cardiovascular system revealed prominent tachycardia, and both heart sounds (S1 and S2) were acoustically distinct on a background of a pan-systolic murmur of grade III intensity, which plateaued before disappearing at diastole. The high-pitched rumbling was audible over the left apex region, corresponding to the fifth left intercostal space, laying adjacent to the mid-clavicular line. In this region of the chest wall, the sound of murmurs can be more easily heard by maneuvering the patient laterally on the left decubitus position with radiation to the axillar region. No other murmurs, including other cardio-adventitious heart sounds, such as S3, S4, opening snaps, ejection clicks, or pericardial rubs, were audible over the tricuspid, aortic, pulmonary, and Erb’s areas of the anterior chest wall.

Examination of the respiratory system revealed symmetrical chest movements that were rapid and consistent with a thoraco-abdominal breathing pattern. Bronchial breath sounds with equal air entry into both lung fields were present on auscultation without the presence of stridor, wheeze, crepitations, or pleural rubs.

Inspection of the abdomen revealed no evidence of an enlarged liver or spleen. The abdomen was soft to palpation without guarding or tenderness in any of the major quadrants, including the posterior flanks. An in situ Foley catheter showed the presence of a small amount of residual urine. Bowel sounds were present and normal.

A central nervous system examination revealed a Glasgow coma scale of 10/15, which represented an eye-opening response only to pain stimulus, inappropriate verbal responses, and motor responses limited to movements of the limbs in response to localized pain stimuli. Muscle tone appeared to be intact with no weakness or flaccidity in the extremities, including normal reflexes and normal plantar response resulting in a negative Babinski sign. Other neurological deficits observed involved cranial nerves II and III. The other cranial nerves (I and IV–VIII) were not assessable except for cranial nerves IX and X, which remained intact. Additionally, lagophthalmos was present in both eyes with exposed corneas. The anterior segment of the right eye had minimal conjunctival injection when compared to the marked conjunctival injection with a corneal infiltrate at the limbal 4 o’clock position (Figure 2) with anterior chamber findings of globular infiltrates and Roth spots.

Bedside ultrasound demonstrated a >50% collapsible inferior vena cava on the longitudinal axis and an oscillating vegetation 30 × 18 mm in the left atrium. Several boluses of intravenous crystalloid had no impact on the low blood pressure, and noradrenalin was started for inotropic cardiac functioning and peripheral vasoconstriction. The finding of a large vegetation was suggestive of IE, and blood was collected by venipuncture from three different sites prior to the administration of targeted empiric antibiotic. Vancomycin was administered up to 1 g/day to maintain the vancomycin trough levels at 20 µg/mL. Further management included transthoracic echocardiography (Figure 3) and hematological and biochemical investigations, as shown in Table 1.

Transthoracic echocardiography demonstrated vegetation in the mitral valve with ruptured chordae tendinae and acute mitral regurgitation. The blood culture was positive for MRSA that was resistant to oxacillin with a vancomycin minimal inhibitory concentration of 1.0 µg/mL (Appendix A).

The present case had clinical findings that met the Dukes criteria (Appendix A). These included two major criteria required for a definitive diagnosis of IE: (i) the presence of a typical microorganism that causes IE and (ii) evidence of endocardial involvement, demonstrated by the presence of vegetation by echocardiography. In addition to the two major criteria, four of the five minor criteria were also met, including (i) a predisposing factor, such as IVUD; (ii) an elevated body temperature above 38 °C; (iii) the presence of a vascular phenomenon, such as Janeway lesions; and (iv) an immunological phenomenon, such as Oslers nodes and Roths spots, which are classic micro-embolic lesions that are considered to be specific to IE but are now fairly rare due to early detection and treatment. The presence of multiple embolic lesions in the present case reflected the chronic course of the disease at the time of presentation.

Additionally, neuroimaging revealed multiple bilateral hyperdense lesions in the occipital region that were suggestive of hemorrhagic transformations (Appendix A). The presented case subsequently required endotracheal intubation and mechanical ventilation with worsening of the consciousness level and was managed in the intensive care unit without any clinical improvement until the clinical trajectory rapidly deteriorated with a fatal irreversible arrhythmia.

## 4. Discussion

### 4.1. MRSA

The resistance of *Staphylococci* to methicillin was first detected in 1961, and since then, several repertoires of genes responsible for virulence factors and resistance to antibiotics, antiseptics, and heavy metals in *Staphylococcus aureus* have been described [21,22]. Over the subsequent decades, MRSA has caused periodic epidemics and is considered to be an emerging nosocomial pathogen, i.e., a cause of infections in healthcare settings, with the potential to cause severe diseases such as bacteremia, pneumonia, endocarditis, and osteomyelitis [23].

### 4.2. Methicillin-Resistant and Other Virulence Factors

The resistance to antibiotics resulted from the horizontal transfer of a mobile genetic element onto *Staphylococcus* species [21]. The gene encoding methicillin resistance (*mecA*), which confers resistance to beta-lactams, has one of four classes of the staphylococcal cassette chromosome *mec* (SCC*mec*), which produces low-affinity penicillin-binding proteins that are abbreviated as PBP2 or PBP2a. In contrast to MRSA strains found in healthcare settings, other virulence factors, such as Panton–Valentine leucocidin (PVL), a pore-forming bacterial leukotoxin that causes the lysis of leukocytes and cell necrosis, were observed to cluster among strains of MRSA outside of healthcare and community settings, with such strains including strains USA100–400 [24]. Nevertheless, the present spillover of strains blurs the distinction between healthcare-associated and community-acquired MRSA infections [25].

### 4.3. MRSA as the Causative Agent Was an Unexpected Finding

In the present case, the blood cultures collected at the time of presentation to the hospital yielded MRSA. To our knowledge, there has been no recent documentation of MRSA in any patient at IGMH prior to the present case, not even among patients with severe diseases such as IE. The reason for this may be that diseases caused by MRSA do not need to be reported, and there is no surveillance system for MRSA within the institution or at a national level. Similarly, MRSA was not among the etiological agents in a retrospective study conducted at IGMH looking at the etiologies of blood stream infections MRSA [26]. Hence, although IE was clinically suspected in the present case, MRSA as the causative agent was an unexpected finding.

### 4.4. This Case Demonstrates the Threat of MRSA

This case demonstrates the threat of MRSA to public health, as the bacterium has the potential to cause detrimental consequences. Furthermore, it is anticipated that similar cases may be encountered in the future. Thus, clinicians should be more vigilant and advocate to public health authorities to implement a surveillance system for blood stream infections.

#### IE Diagnosis of the Present Case

The diagnosis of IE primarily involves correctly identifying the etiological agent through serial blood cultures and the use of molecular techniques to identify fastidious pathogens [27]. Conventionally, it is recommended to collect venous blood for culture in three separate culture sets, at three different sites, and at three periodic intervals. This helps to determine the persistence of the bacteremia, which is vital for the diagnosis and management of the disease. In the present case, MRSA was identified in all blood samples collected at three time points, which reflected the protracted bacteremia.

### 4.5. The Maldives Situation

The Maldives is a small-island developing state located in the Indian Ocean, where tropical diseases such as dengue, chikungunya, Zika, and scrub typhus are endemic [28,29,30,31,32]. It is the smallest country in Asia, with a population of 557,426 people and with a median age of 28 years and an overall life expectancy of 79.61 years. (2020 est.). The health sector is comprised of one hundred and sixty-four primary health centers, thirteen atoll hospitals, six regional hospitals, and five hospitals in the capital city. The main economical industry is tourism, attracting almost two million travelers annually. Although both the literacy rate (97.7%) and GDP per capita (USD 8280) are comparably highest in the region, unemployment and poverty represent 6.2% and 8.2% of the population, respectively. Difficult socio-economic circumstances are considered to contribute to social vices such as drug use.

Small-island developing states such as the Maldives face an increasing rate of drug abuse. The United Nations Office of Drug and Crime reported from a survey conducted in 2013 across the atolls that 22.9% of the 13,714 respondents were active drug users, of which 2.3% were identified as IVDU. Further, a wastewater-based epidemiological study conducted in 2019 demonstrated the presence of multiple illicit substances, with detection of opioid quantities comparably higher than the other parts in the region [33].

### 4.6. IVDU and Other Predisposing Factors

In cases with IVDU, the characteristic cardiac involvement is involvement of the right-sided heart valves, which increases the risk of embolization to the lungs, causing pulmonary embolism [34]. Conversely, in the present case, the vegetation was located on the mitral valve without any involvement of the valves on the right side of the heart. Hence, we considered that the pathophysiology of IE in the present case may not have been directly due to IVDU but was rather indirectly due to the introduction of MRSA through unsterile injection techniques. Additional predisposing factors, such as a congenital pathology of the mitral valve leaflets, may have been present but undetectable in the present case.

#### Comparison with Other Reported Cases

We also compared other case reports that match the case presented in Appendix A [35,36]. At presentation to the hospital, the presented case with IE was in a critical condition affecting the valve on the left side of the heart. Involvement of the mitral and aortic valves poses a much greater risk of severe complications, leading to a poor prognosis and reduced survival after embolic events causing infarcts in multiple organs. In the present case, embolization to the brain was observed at multiple locations, including septic seeding to the eye. Due to the severe disease at the time of presentation and reduced GCS suggestive of raised intracranial pressure, early endovascular interventions were not performed.

## 5. Conclusions

We described the clinical findings of a fatal case of IE. Additionally, this report showcases important clinical findings that are pathognomonic to infective endocarditis, which we anticipate will be highly useful for clinicians and medical students. Early recognition of the disease is essential to avoid complications and fatal outcomes. Although the isolated MRSA strain was still sensitive to several antimicrobials, which could be used in salvage therapy, proper preventative measures, such as hand hygiene and contact precautions, are pivotal and fundamental to control and contain the spread of MRSA.

## Figures and Tables

**Figure 1 tropicalmed-08-00053-f001:**
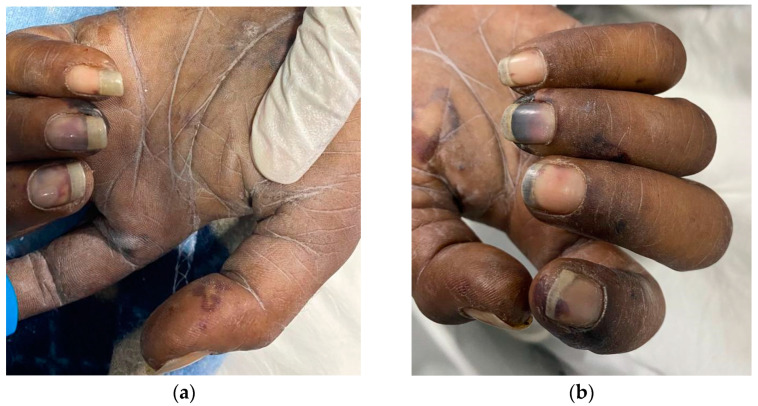
Peripheral stigmata of IE. (**a**,**b**) Both the left and right hands showed multiple Janeway lesions and splinter hemorrhages.

**Figure 2 tropicalmed-08-00053-f002:**
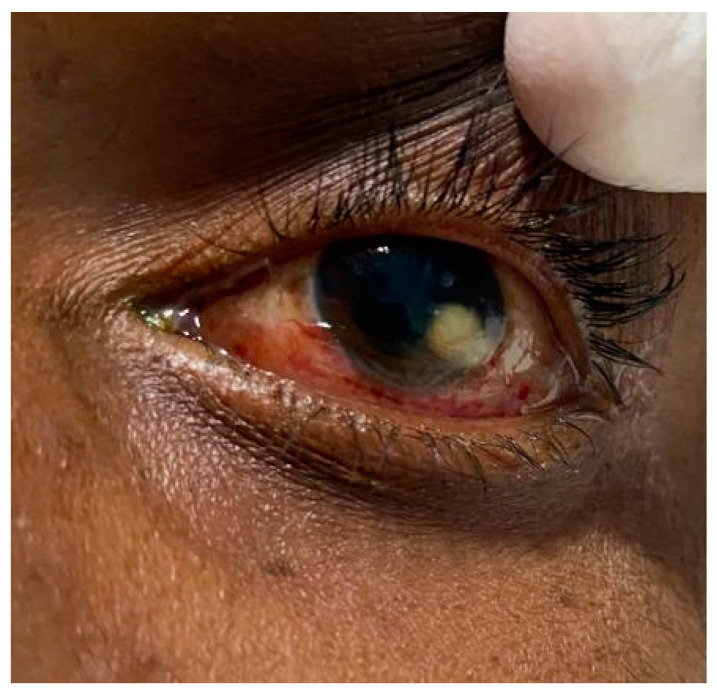
A hypopyon caused by septic embolization in the left eye that was suggestive of uveitis.

**Figure 3 tropicalmed-08-00053-f003:**
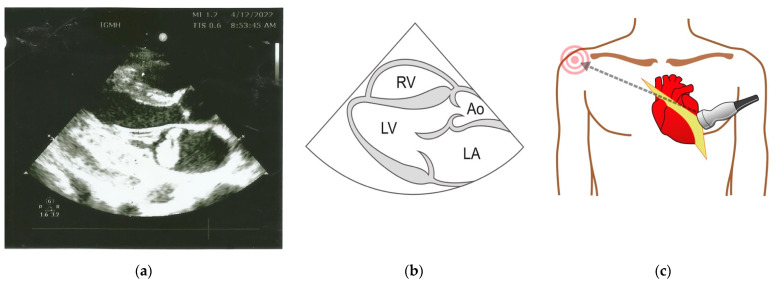
(**a**) Two-dimensional echocardiogram showing a large echogenic mass in the left atrium. (**b**) Normal anatomical landmarks seen in the long axis view (RV, right ventricle; LV, left ventricle; LA, left atrium; Ao, aorta). (**c**) Parasternal position of the probe for transthoracic echocardiography.

**Table 1 tropicalmed-08-00053-t001:** Laboratory investigations performed over the course of the illness.

Day of Illness (Days)	Day 4 ^†^	Day 5	Day 6	Day 7	Day 8	Day 9	Day 10	Day 12
Hospitalization ^‡^ (Days)		Day 1	Day 2	Day 3	Day 4	Day 5	Day 6	Day 7
Leukocytes (5000–10,000/µL)	13,700	21,000	21,600	20,800	18,500	26,000	25,000	48,100
Neutrophils (40–60%)	79.1	68.0	82.8	76.7	74.8	77.8	77.7	78.7
Lymphocytes (12.2–47.1%)	3.4	14.0	6.3	12.6	16.1	11.0	12.8	7.4
Eosinophils (0.0–4.4%)	0.9	0	0.2	0	0	0.1	0.3	0.1
Basophils (0.0–0.7%)	0.5	0	0	0	0	0	0	0
Monocytes (4.4–12.3%)	16.1	17	6.4	6.3	5.4	3.9	3.9	4.6
Hemoglobin (11.9–15.4 g/dL)	13.1	12.7	10.5	9.9	9.7	9.2	8.0	7.9
Hematocrit (36.2–46.3%)	38.6	39.9	32.4	28.8	29.5	24.9	24.0	23.2
Platelets (151,000–304,000/µL)	23,000	4700	3500	37,300	33,400	26,600	25,000	29,000
Creatinine (0.7–1.2 mg/dL)	2.67	3.13	1.95	1.26	1.21	1.09	1.19	1.07
Urea (19.0–44.1 mg/dL)	209.0	258.9	222.5	192.6	175.4	158.3	145.5	115.5
Sodium (136–145 mmol/L)	138	142	150	157	153	151	152	148
Potassium (3.5–5.1 mmol/L)	3.6	4.0	3.5	3.3	4.9	4.1	4.0	4.2
Total bilirubin (0.2–1.2 mg/dL)		3.6	2.1	1.4	0.8	1.0	1.3	1.3
Direct bilirubin (0.0–0.5 mg/dL)		2.3	1.4	0.9	0.3	0.4	0.8	0.9
Albumin (35–5.2 g/dL)		2.6	2.1	2.2	2.2	2.1	2.0	1.9
Protein (6.4–8.3 g/dL)		6.0	5.0	5.1	5.7	5.7	5.6	6.2
Aspartate aminotransferase (5.0–34.0 IU/L)		104	124	63	38	26	45	85
Alanine aminotransferase (0.0–55.0 IU/L)		49	52	37	31	25	36	71
Alkaline phosphatase (40.0–150.0 IU/L)		78	58	69	61	79	94	223
INR (2.0–3.0)		1.2	1.3	1.1		1.1		1.3
APPT (24.6–38.8 s)		37.6	41.7	38.2		38.9		42.6
PT (11.1–13.1 s)		14.6	15.8	13.0		13.7		16.5
CRP (0.0–0.5 mg/dL)	45.5	43.4	46.5	39.0	30.7	25.3	26.8	29.2
CK (30–200 IU/L)		382	476					
CKMB (0.0–24.0 IU/L)		25	51					
LDH (140–280 IU/L)		681						
Troponin I (0.0–0.3 ng/mL)		3.6						
Lactate (4.5–19.8 mg/dL)		69						
Hepatitis B surface Ag		Positive						
Anti-hepatitis B surface Ab		Positive						
Anti-hepatitis C Ab		Negative						
RPR		Non-reactive						
Anti-HIV Ag/Ab		Negative						
SARS-CoV-2 RT-PCR		Negative						
Blood culture		Positive		Positive				

^†^ Day 4: investigations performed at primary health center. ^‡^ Hospitalization: the case presented was transferred from the emergency department (IGMH) to the intensive care unit at the National Cardiac Center within three hours of presentation. INR, international normalization ratio; APPT, activated partial thromboplastin time; PT, prothrombin time; CRP, c-reactive protein; CK, creatinine kinase; CKMB, creatinine kinase myoglobin binding; LDH, lactate dehydrogenase; RPR, rapid plasma regain; Ag, antigen; Ab, antibody; RT-PCR, real-time polymerase chain reaction.

## Data Availability

The data presented in this study are available on request from the corresponding author. The data are not publicly available to ensure the privacy of the study participant.

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
