# Peer review of "A Fatal Case of Native Valve Endocarditis with Multiple Embolic Phenomena and Invasive Methicillin-Resistant Staphylococcus aureus Bacteremia: A Case Report from the Maldives"

_tropicalmed, 2023, doi:10.3390/tropicalmed8010053_

Round 1
Reviewer 1 Report
Please see the attached document for comments.

Author Response
We thank the reviewer for the valuable time and welcome the excellent feedback.
Please find below our responses to the reviewer's comments and the changes made in the revised manuscript.
- Lines 18,20, 109: The name of the hospital is probably not Indhira but should be Indira. Please check and correct at all places in the manuscript.
>>We are thankful for bringing this to our attention. In the revised manuscript, we have corrected the misspelled institution name from “Indhira” to the correct spelling “Indira” in all places in the manuscript.
- Please include some statistics on infectious diseases, especially endocarditis, in the Maldives. In addition, a general background of the healthcare system, and the number of peripheral and tertiary care hospitals in Maldives will be helpful. Lastly, the socio-economic profile of people in the Maldives will also strengthen the paper.
>> We appreciate this suggestion. In the revised manuscript, we have addressed the reviewer's suggestion and included a sentence about infectious diseases in the Maldives (Lines 480-482), in particular, bloodstream infections at IGMH (Lines 436-438) and its healthcare system and the socio-economic profile of the people in the Maldives (Line 482-490).
“The Maldives is a small island developing state located in the Indian Ocean where tropical diseases like dengue, chikungunya, Zika, scrub typhus, typhoid, and tuberculosis are endemic.”
“Similarly, MRSA was not among the etiological agents in a retrospective study conducted at IGMH looking at the etiologies of bloodstream infections MRSA.”
“It is the smallest country in Asia with a population of 557,426 people with a median age of 28 years and an overall life expectancy of 79.61 years. (2020 est.). The health sector comprises of one hundred and seventy-two primary health centers, thirteen atoll hospitals, six regional hospitals, and five hospitals in the capital city. The main economical industry is tourism, attracting almost two million travelers annually. Although both the literacy rate (97.7%) and GDP per capita (USD 8,280) are comparably highest in the region. Unemployment and poverty represent 6.2% and 8.2% of the population respectively. Difficult socio-economic circumstances are considered to contribute to social vices such as drug use.”
- Toward the end of the introduction, the authors should be able to establish the need for describing this case study. That need is missing in the manuscript.
>>We are thankful to the reviewer for this suggestion. In the revised manuscript, in the Introduction, we included an additional paragraph (Lines 117-126) describing the rationale and importance of describing the case presented.
“This study describes a severe complex medical condition requiring a multi-disciplinary approach for effective treatment and management required in order to prevent the patient from succumbing to illness. Further, we highlight the alarming discovery of multi-drug resistant bacteria such as MRSA within the community in the Maldives and bring forth worsening social issues and behaviors which risk acquiring and transmitting pathogens within certain groups of individuals in the Maldivian population. Additionally, this study will showcase important clinical findings that are pathognomonic to infective endocarditis which we anticipate would be highly useful for clinicians and medical students.”
- Lines 103-105 – Data are 10 years old from 2013. Given the striking changes in drug use data in most countries, I would strongly encourage the authors to update the data.
>> We are thankful for this suggestion. In the revised manuscript, we have included additional information reflecting the extent of drug use in the Maldives. In addition, we have added the following sentence in (Lines 495-496) in the revised manuscript.
“Further, a wastewater-based epidemiological study conducted in 2019 demonstrated the presence of multiple illicit substances with detection of opioid quantities comparably higher than other parts in the region.21”
- Lines 106-107: The authors state a case study in adolescence but then, in line 121, the authors state the case of a 31-year-old adult. Please correct.
>> We thank the reviewer for bringing this to our attention. In the revised manuscript, we have deleted the complete sentence. (Lines 106-107)
- Line 122: It would be helpful to describe a peripheral health center since readers in other countries and settings might not be familiar (e.g., in the U.S., several such cases come through the Emergency Department).
>> We appreciate this suggestion. For reader’s clarity, in the revised manuscript, we have rephrased “peripheral health center” to primary health center. (Line 163)
- Line 123: Was the patient brought to the ED at the central referral tertiary care hospital? Please make that clear.
>>We appreciate this suggestion. In the revised manuscript, we have revised the sentence for reader clarity. (Line 162-164)
“A 31-year-old male with a provisional diagnosis of sepsis was promptly medically evacuated from a primary health center to the emergency department at a central referral tertiary care hospital (IGMH) in the Maldives.”
- Lines 282-296: Move this paragraph on Duke criteria in the main case report section, instead of Discussion. A short table showing the Duke criteria applicable to this case for diagnosis of endocarditis could provide an instant view.
>> As suggested by the reviewer, in the revised manuscript, we have included the Dukes criteria in the Case Description (Lines 322-332). We have also included a short table showing the Duke criteria in the supplementary material as Table S2.
“The present case had clinical findings that met the two major criteria required for a definitive diagnosis of IE: i) the presence of a typical microorganism that causes IE, and ii) evidence of endocardial involvement, demonstrated by the presence of vegetation by echocardiography. In addition to the two major criteria, four of the five minor criteria were also met, including i) a predisposing factor, such as IVUD, ii) an elevated body temperature above 38oC, iii) the presence of a vascular phenomenon, such as Janeway lesions, and iv) an immunological phenomenon, such as Osler's nodes and Roths spots, which are classic micro-embolic lesions that are considered to be specific to IE but are now fairly rare due to early detection and treatment. The presence of multiple embolic lesions in the present case reflected the chronic course of the disease at the time of presentation.”
Table S2. Modified Dukes criteria for infective endocarditis
|
Major criteria |
Case presented |
|
Supportive laboratory evidencea |
Positive blood cultures for MRSA |
|
Evidence of endocardial involvementb |
Vegetation on mitral valve |
|
Minor criteria |
|
|
Predisposing heart condition or IVDU |
Active IVDU |
|
Fever > 38oC |
Febrile at presentation |
|
Vascular phenomenonc |
Multiple Janeway lesions present |
|
Immunological phenomenond |
Osler’s nodes and Roths spots present |
|
Positive blood culture not meeting the Major criterione |
- |
a Typical microorganism of infective endocarditis from two separate blood cultures: Viridians streptococci, Staphylococcus aureus, Streptococcus bovis, HACEK group or community-acquired enterococci, in the absence of a primary focus. Additional a single positive culture for Coxiella brunetii or serology (antibody titer >1:800
b Echocardiogram supportive of infective endocarditis. Definition of positive findings: Oscillating intracardiac mass, on the valve or supporting structures, or in the path of regurgitation jets, or on implanted material, in the absence of an alternative anatomic explanation or myocardial abscess or new partial dehiscence of prosthetic valves. New valvular regurgitation.
c Major arterial emboli, septic pulmonary infarcts, mycotic aneurysm, intracranial hemorrhage, conjunctival hemorrhage, and Janeway lesions.
d Glomerulonephritis, Osler’s nodes, Roths spots, rheumatoid factor
e Excluding single positive cultures for coagulase-negative staphylococci and organisms that do not cause endocarditis or serological evidence of active infection with organisms consistent with infective endocarditis
IVDU: intravenous drug use, MRSA: methicillin-resistant Staphylococcus aureus
This table was obtained from a textbook. “Infective endocarditis: a multidisciplinary approach. ISBN: 978-0-12-820657-7
- Please show the timeline of the patient from the peripheral hospital to IGMH to the National Cardiac Center; e.g., what was the length of stay at each of the three facilities?
>>>> We appreciate this suggestion by the reviewer. In the revised manuscript, we have included a footnote in Table 1 describing the timeline the case presented stayed in each facility.
Ҡ Day 4: investigations performed at the primary health center.
‡ Hospitalization: the case presented was transferred from the emergency department (IGMH) to the intensive care unit at the National Cardiac Center within three hours of presentation.”
- Was this the first episode of IE for the patient?
>> Yes, there was no prior diagnosis of IE.
- Lines 302-303: The authors have conjectured the IE may not have due to IVDU. Was the patient tested for drugs? Which ones?
>>>> In the case presented, a urine sample was sent for toxicology screening (barbiturates, amphetamines, benzodiazepine, cocaine, methamphetamine, morphine, methadone, phencyclidine, methylenedioxymethamphetamine, tetrahydrocannabinol)
Unfortunately, due to the retrospective nature of this report we were unable to locate the results of the toxicology report as they were missing from the medical chart. Further, the toxicology screening was outsourced; it was not in the hospital's electronic data system.
- Was MRSA the only organism identified?
>>Yes, MRSA was the only organism identified by the automated system. Further, for readers' clarity, we have included an additional paragraph describing how IVDU are exposed to other bacteria. (Lines 396-399)
“Cleaning the paraphernalia by licking exposes IVDU to the HACEK group of bacteria and Pseudomonas aerugionas through environmental contamination of methods employed in he preparation of the illicit substance”
- Discussion should include how this case study compares with other case studies in literature.
>> We thank the reviewer for this suggestion and as suggested we performed a brief review for cases to match with the case presented. In brief, we used the google scholar search engine to match the presented case with other similar IE cases. A total of five keywords (native valve; Infective endocarditis; MRSA; IVDU; Septic emboli; and Case reports) along with a three-year period were used. Subsequently, 50 papers published in the English language were retrieved for further scanned. After reading the abstracts, of which, we first excluded 46 papers. One was the current study in pre-print form, one was duplicated, 10 were either general reviews or basic literature, or book chapters for basic etiology, diagnosis, and treatment of IE, 11 were analyses of a case series either in retrospective or prospective approach, three case reports provided the information regarding IE with MSSA, and remaining 20 were not relevant with the presented case as likewise, they described different areas of interest such as bioengineering, cardiac magnetic resonance imaging, or general infectious diseases. This exclusion made a total of four papers to be included in a detailed review. Among them, next, another two were repudiated because one paper lacked the details on the patient’s presentation and cardiovascular-related findings, and another reported about a patient with underlying malignancy and ongoing treatment for many years. Consequently, two final case reports were eventually included in our review and we have tabulated the findings for easy comparison in the supplementary section in Table S3.
|
|
Present case |
Other cases |
|
|
Case-1a |
Case-2b |
||
|
General characteristics |
· 31-year-old male from the Maldives · IVDU (+) · No prior history of IE |
· 26-year-old male from USA · IVDU (+) · No prior history of IE |
· 36-year-old female from USA · IVDU (+) · No prior history of IE |
|
Presenting complaints |
· Referred as a case of septicemia (Adjunct with a four days’ history of fever followed by decreased conscious level) |
· Fever and cough for a week |
· Fever and cough for two weeks |
|
Gasglow Coma Scale |
· 10/15 |
15/15 |
Not provided |
|
Vitals at presentation |
· Temperature: 38.1o C · Pulse: 142 beats per min. · Blood pressure: 88/58 mmHg · Respiratory rate: 52 breaths per min. |
· Hemodynamically stable
|
· Pulse: 124 beats per min. · Blood pressure: 112/59 mmHg · Respiratory rate: 27 breaths per min.
|
|
Stigmata of infective endocarditis |
Janeway lesion Osler’s nodes Splinter hemorrhages |
Janeway lesion Osler’s nodes
|
Not present |
|
Cardiovascular examinations findings |
· Tachycardia · Pansystolic murmur (Grade-3 intensity) |
· Regular rate and sinus rhythm · No murmur |
· Holo systolic murmur |
|
Echocardiographic findings |
· Vegetation (30 x 18mm) on mitral valve · Ruptured chordae tendinae Mitral regurgitation |
· Vegetation (15 x 13mm) on tricuspid valve |
· Vegetation (18 x 24mm) on tricuspid valve |
|
Hematological and inflammatory marker |
· Leukocytosis with thrombocytopenia · Elevated CRP |
· Not provided |
· Leukocytosis |
|
Isolated micro-organism from blood |
MRSA |
MRSA |
MRSA |
|
Other complications |
· Septic embolization to brain, eye and extremities
|
· Septic embolization to lungs, spleen, kidneys, prostate, bladder, and extremities
|
· Septic embolization to lungs |
|
Case mangement |
· Inotropic support · Mechanical ventilation · Vancomycin 20mcg/mL |
· Excision of mitral valve left leaflet vegetation’s and repair of mitral valve perforation. · Excision of the tricuspid valve chordal vegetation and repair of tricuspid valve chordal rupture · Repair of mitral valve perforation · Vancomycin, piperacillin-tazobactam, daptomycin
|
· Vancomycin, piperacillin-tazobactam, clindamycin · Percutaneous extraction of the tricuspid valve vegetation using suction filtration and veno-venous bypass · Aspiration of the right knee
|
|
Outcome |
· Demise |
· Discharge after six weeks |
· Discharge after six weeks and lost to follow-up |
- Reorganizing the entire paper with subheadings (or starting each paragraph with a theme in the first sentence) will greatly strengthen the readability and flow of the paper.
>> We thank the reviewer for this suggestion and as suggested, we have reorganized the manuscript with subheadings.
- Several of the references are very old. There has been enormous research done on endocarditis in the past 5 years. Please use updated references.
>> As suggested, in the revised manuscript, the references have been updated.
- Lastly, please state how this case study is going to be useful for clinicians and/or patients.
>> As suggested, in the revised manuscript, we have included additional sentences highlighting the importance of this case to clinicians and students. (Lines 441-443)
“Additionally, this report showcase important clinical findings that are pathognomonic to infective endocarditis which we anticipate would be highly useful for clinicians and medical students.”

Reviewer 2 Report
In this article, the authors present a clinical case of rapidly fatal bacterial endocarditis caused by a methicillin-resistant Staffolococcus Aureus in a drug addict. The presentation of the case is thorough. However, this article should be shortened by at least one third, especially reducing the introductory part (far too long and with redundant historical references)
Specific comments.
Lines 106-107. Authors state: “Here, we present a fatal case of native valve IE due to intravenous drug use in adolescence.” However, later on (line 121) the case of a 31-year-old man (certainly not an adolescent) is described.
Author Response
We thank the reviewers for their valuable time.
According to the reviewer's suggestion, we have deleted the introduction as much as possible, but this is the limit since we specifically targeted this case report to medical students and clinical researchers, and this manuscript will be considered academic teaching material in our institutions. Hence, it would be more appealing and enhance the reader's interest if the writing style could link the past to the present situation. The authors feel strongly that even though some of the discoveries may be outdated, information about these medical milestones will help readers better understand endocarditis.
We would like the Reviewer and the Editor to consider our request not to shorten the manuscript further. Lastly, one of the reasons we are fond of the Journal of Tropical Medicine and Infectious Diseases is that there is no word limit.
Specific comments.
Lines 106-107. Authors state: “Here, we present a fatal case of native valve IE due to intravenous drug use in adolescence.” However, later on (line 121) the case of a 31-year-old man (certainly not an adolescent) is described.
>> We are thankful to both reviewers for bringing forth this error. In the revised manuscript, we have corrected this error.
Round 2
Reviewer 1 Report
Thank you for revising the manuscript based on the feedback. While I can understand the extra effort involved in the revision, I think the paper is much stronger and clearer now.
Author Response
Thank you for the time and consideration.
Reviewer 2 Report
I understand the authors' comments. In my opinion, however, this article is still too long for a clinical case presentation.
Author Response
As suggested by the reviewer and academic editor, we have shortened the manuscript.